# High-Temperature Oxidation Resistance of Fe-Free AlCoCrNiNb_0.2_ and AlCoCr_0.5_NiNb_0.2_ High-Entropy Alloys

**DOI:** 10.3390/ma18153701

**Published:** 2025-08-06

**Authors:** Olga Samoilova, Svetlana Pratskova, Nataliya Shaburova, Ahmad Ostovari Moghaddam, Evgeny Trofimov

**Affiliations:** 1Department of Materials Science, Physical and Chemical Properties of Materials, South Ural State University, 76 Lenin Av., 454080 Chelyabinsk, Russia; shaburovana@susu.ru (N.S.); trofimovea@susu.ru (E.T.); 2Research and Innovation Services, South Ural State University, 76 Lenin Av., 454080 Chelyabinsk, Russia; se_pratskova@mail.ru; 3Department of Analytical and Physical Chemistry, Chelyabinsk State University, 129 Bratiev Kashirinyh Street, 454001 Chelyabinsk, Russia; 4Department of Applied Mathematics, National Research University Higher School of Economics, 101000 Moscow, Russia; mostovari@hse.ru

**Keywords:** high-entropy alloys, microstructure, alloying, niobium, high-temperature oxidation resistance, oxide film

## Abstract

The microstructure, phase composition, and high-temperature oxidation resistance of Fe-free AlCoCrNiNb_0.2_ and AlCoCr_0.5_NiNb_0.2_ high-entropy alloys (HEAs) were investigated. In the as-cast HEAs, niobium was found to mainly release as a Laves phase in the interdendritic region, and its solubility in the dendrites of the BCC solid solution was about 2 at.%. Both samples exhibited parabolic behavior during 100 h oxidation at 1000 °C and 1100 °C. The AlCoCrNiNb_0.2_ alloy demonstrated higher resistance to high-temperature oxidation compared to AlCoCr_0.5_NiNb_0.2_. The specific weight changes after 100 h of isothermal holding at 1000 °C and 1100 °C were 0.65 mg/cm^2^ and 1.31 mg/cm^2^, respectively, which are superior compared to the Fe-containing HEAs. Cr was revealed to play an important role in the oxidation behavior of the HEAs, decreasing the parabolic oxidation rate constant and increasing the activation energy of the oxidation process in the alloys.

## 1. Introduction

High-entropy alloys (HEAs) are multicomponent alloys composed of five or more elements in an equiatomic (or close to it) ratio to achieve the maximum configurational entropy of mixing. This unusual approach usually provides these multicomponent alloys with improved characteristics compared to conventional alloys. Al*_x_*CoCrFeNi-type HEAs exhibit an ideal unification of mechanical properties like increased strength, toughness, ductility, reduced brittleness, etc. [1,2], which is accompanied by improved tribological characteristics [3,4,5] and high corrosion resistance [6,7,8]. Moreover, further studies have shown the potential applications of HEAs as heat-resistant materials [9,10,11], including the bond coat in thermal barrier coatings [12,13,14,15].

Currently, a major research focus is on studying the effects of additional alloying elements on the properties of the Al*_x_*CoCrFeNi-based HEAs, particularly in the context of enhancing resistance to high-temperature oxidation. Several studies have reported that the high-temperature oxidation resistance of alloys can be significantly improved by introducing elements such as Y/Hf [16], Y/Ta/Hf [17], Y [18], or Pt [19]. The positive effect of yttrium and platinum introducing these elements is primarily due to their influence on the diffusion processes occurring during oxidation. Further, the presence of these elements provokes the formation of an aluminum-oxide protective film, while it slows down the oxidation processes of other components in the alloy.

Among the promising alloying elements, Nb has attracted particular attention [20,21]. For example, Yang et al. [20] demonstrated that the addition of Nb to Al_0.2_Co_1.5_CrFeNi_1.5_Ti_0.3_ HEA reduced the specific weight gain to half after 50 h oxidation at 900 °C. Similarly, Wu et al. [21] reported that the CoCrFeNiMo_0.2_Nb_0.2_ HEA, after isothermal exposure in air at 1000 °C for 60 h, exhibited a nearly 50% reduction in specific weight gain compared to the niobium-free alloy. In addition to improving oxidation resistance, Nb has been shown to positively influence mechanical and tribological properties. Feng et al. [22] found that alloying CoCrFeMnNi with niobium (CoCrFeMnNiNb*_x_*) enhanced several mechanical- and wear-related characteristics. These findings are consistent with those of Malatji et al. [23], who observed that Nb-containing alloys exhibited higher hardness, compressive strength, and wear resistance compared to the Nb-free HEAs.

A promising research direction involves the investigation of the properties of Al*_x_*CoCrNiMeᵧ-type HEAs, where Me denotes an additional alloying element and iron is intentionally excluded due to its detrimental impact on alloy performance. For instance, Pan et al. [24] reported excellent corrosion resistance of the Fe-free AlCoCrNiNb*_x_* alloy in a 3.5 wt.% NaCl solution. Similarly, Gawel et al. [25] demonstrated that the Al_20_Co_25_Cr_25_Ni_25_Si_5_ composition exhibits enhanced high-temperature oxidation resistance compared to its iron-containing counterparts.

The objective of the present study is to evaluate the high-temperature oxidation resistance of two newly developed Fe-free AlCoCrNiNb_0.2_ and AlCoCr_0.5_NiNb_0.2_ high-entropy alloys. In contrast to the HEA compositions prepared and studied elsewhere [24], the present study investigates alloys with different Cr/Nb and Al/Cr ratios. The compositions proposed HEAs here were synthesized for the first time. Further, there is no work devoted to the study of high-temperature oxidation of Fe-free Nb-containing HEAs that ensures the scientific novelty of this research. This work will focus on determining the oxidation kinetics, examining the morphology and composition of the resulting oxide scales, and assessing their continuity and protective properties.

## 2. Materials and Methods

The HEAs ingots were produced by melting granules and powders of high-purity metals (>99.9 wt.%) using the vacuum furnace NABERTHERM VHT 8/22-GR (Nabertherm GmbH, Lilienthal, Germany). The mixture of materials was loaded into alumina crucibles with a lid and melted; during the melting process, the temperature in the furnace interior reached 1670–1730 °C. Three remeltings were carried out to obtain a homogeneous composition. The chemical composition of the resulting ingots was controlled using an OPTIMA 2100 DV inductively coupled plasma atomic emission spectrometer (Perkin Elmer, Perten, Waltham, MA, USA).

The oxidation experiments were conducted in a Plavka.Pro PM-1 SmartKiln muffle furnace (Plavka.Pro, Korolev, Russia) at temperatures of 1000 °C and 1100 °C for a duration of 100 h in air. The procedure adopted to test the samples is described in detail in our previous work [26]. The test temperatures were selected based on the presumption that these Nb-containing HEAs could be good candidates for metallic bond coat in thermal barrier coatings as a replacement for nickel superalloys. The use of HEAs as a bond coat implies working temperatures in the range of 1000 °C and above, reaching 1100 °C and even 1150–1200 °C.

To study the microstructure before and after oxidation, as well as to determine the thickness of the formed oxide film, polished sections were made from the experimental samples. For this, a Delta AbrasiMet (Buehler, Leinfelden-Echterdingen, Germany) cutting machine, a SimpliMet 1000 (Buehler, Leinfelden-Echterdingen, Germany) pressing machine, and an EcoMet 250/AutoMet 250 (Buehler, Leinfelden-Echterdingen, Germany) grinding machine were used. Grinding was carried out sequentially on 250 μm, 75 μm, 25 μm, and 9 μm sandpaper, and a 3 μm diamond suspension was used for final polishing.

Microstructural analysis was performed on a JEOL JSM-7001F scanning electron microscope (SEM) (JEOL, Tokyo, Japan). Chemical analysis of microstructural components was carried out using the Oxford INCA X-max 80 energy-dispersive X-ray spectroscopy (EDS) detector (Oxford Instruments, Abingdon, UK). X-ray diffraction (XRD) was carried out on a Rigaku Ultima IV X-ray diffractometer (Rigaku, Tokyo, Japan) using Cu-Kα radiation (λ = 0.15406 Å) with the following shooting parameters: The scanning range varied from 5° to 100°, the step size was equal to 0.02°, and the scanning rate was 5 degrees per minute.

## 3. Results and Discussion

### 3.1. Microstructure and Phase Composition of the As-Cast HEAs

Both as-cast HEAs demonstrated a clearly dendritic microstructure. According to the EDS mapping (Figure 1) and EDS chemical analysis (Table 1), the dendrites (D, dark gray areas) are enriched in Al and Ni; and in the interdendritic (ID) region, there are two regions, one enriched in Cr (ID1, light gray areas), and the other containing significant concentrations of Nb (ID2, white areas). At the same time, Co is almost uniformly distributed over the microstructural components; however, a slightly increased concentration of Co can be seen in ID1 and ID2.

The segregation of niobium into the interdendritic region in a number of HEAs is described by Wu et al. [21], Pan et al. [24], and Jiang et al. [27]. In addition, Ma and Zhang [28] indicated a hypoeutectic microstructure for AlCoCrFeNb*_x_*Ni HEAs with a niobium content of *x* ≤ 0.6.

According to the obtained diffraction patterns (Figure 2), the main phase in both HEAs is a BCC solid solution along with some secondary Laves phase (Co_2_Nb-type). Taking into account the chemical composition of the microstructural components (Table 1), it can be assumed that the Laves phase has a more complex formula, for example, (Co,Cr)_2_Nb. The formation of Laves phases with the introduction of niobium into HEAs was also discovered earlier [21,22,24,27,28,29]. The formation of a BCC solid solution has been also demonstrated by Pan et al. [24] for AlCoCrNi and AlCoCrNiNb*_x_* HEAs. Moreover, the FCC to BCC phase transition with increasing aluminum content from *x* = 0 to *x* = 1 in AlCoCrNi-base HEAs is a well-known phenomenon described in several works [30,31].

The configurational entropy of mixing in alloys can be determined according to [32].(1)$$\Delta S_{\mathrm{mix}}=-R\sum_{i=1}^{n}X_{i}\ln X_{i}$$
where *R* is the universal gas constant (*R* = 8.314 J∙mol^−1^∙K^−1^) and *X* is the atomic fraction of the element in the alloy. When Δ*S*_mix_ ≥ 1.5*R*, the alloy can be considered as high-entropy. For the AlCoCrNiNb_0.2_ alloy, this value is 1.51*R*, and for the AlCoCr_0.5_NiNb_0.2_ composition, the configurational entropy of mixing is 1.50*R*. Thus, both compositions formally meet the criterion of a high-entropy alloy.

### 3.2. High-Temperature Oxidation

The changes in specific weight gain as a function of isothermal holding time are shown in Figure 3. Each point on the graphs represents the result of averaging three parallel experiments; the calculated errors did not exceed 5%. Compared to AlCoCr_0.5_NiNb_0.2_, AlCoCrNiNb_0.2_ HEA demonstrated higher resistance to high-temperature oxidation. For this HEA, the specific weight gain at 1000 °C was 0.65 mg/cm^2^, and at 1100 °C this value was 1.31 mg/cm^2^. The AlCoCr_0.5_NiNb_0.2_ alloy showed weight gains of 1.03 mg/cm^2^ and 2.1 mg/cm^2^ at 1000 °C and 1100 °C, respectively. Our obtained data can be compared with the data of Butler and Weaver [11], who report values of 0.5 mg/cm^2^ and 1.1 mg/cm^2^ after 100 h isothermal holding at 1050 °C for Al_30_(NiCrCoFe)_70_ and Al_12_(NiCrCoFe)_88_ alloys, respectively. Wu et al. in [21] found 2.13 mg/cm^2^ and 1.11 mg/cm^2^ after 60 h of isothermal holding at 1000 °C for the compositions CoCrFeNi and CoCrFeNiMo_0.2_Nb_0.2_, respectively. Our results are also comparable with the value of 0.72 mg/cm^2^ for the AlCoCrFeNi HEA after 50 h holding at 1000 °C reported by Dąbrowa et al. [33]; and they are comparable with those reported by Chen et al. [34], where they obtained weight gains of 1 mg/cm^2^ for Al_0.6_CrFeCoNi and 1.12 mg/cm^2^ for Al_0.6_CrFeCoNiSi_0.3_ HEA under the same experimental conditions. Moreover, both alloys studied in this work exhibited superior oxidation resistance compared to those reported by Zhu et al. [35] for AlCoCrFeNi HEA, where the authors obtained a value of 2.5 mg/cm^2^ after 100 h isothermal holding at 1000 °C. Thus, the Fe-free alloys studied in this work show good resistance to high-temperature oxidation.

The appearance of the curves in Figure 3 corresponds to the parabolic law of oxidation. Specific weight change (*W*) in parabolic law can be related to time as follows:(2)$$W=k\sqrt{\tau }$$
or(3)$${\left(\frac{\Delta m}{A}\right)}^{2}=k_{p}\tau$$
where Δ*m* is mass change (g), *A* is the surface area (cm^2^), *k_p_* is the parabolic oxidation rate constant (g^2^/cm^4^s), and *τ* is the holding time (s).

High-temperature oxidation resistance can also be evaluated by determining the activation energy of the oxidation process. A higher activation energy generally indicates greater resistance to oxidation. The activation energy for oxidation at two different temperatures can be calculated using Equation (4).(4)$$\ln\frac{k_{2}}{k_{1}}=\frac{E_{a}}{R}\left(\frac{1}{T_{1}}-\frac{1}{T_{2}}\right)$$
where *k* is the oxidation rate constant (g^2^/cm^4^s), *E*_a_ is activation energy (J/mole), *R* is the universal gas constant (*R* = 8.314 J∙mole^−1^∙K^−1^), and *T* is temperature (K).

The results of the calculations of the oxidation parameters are summarized in Table 2. Overall, analysis of the oxidation kinetics indicates that the newly developed Fe-free alloys exhibit promising potential for application as heat-resistant materials.

### 3.3. Morphology and Phase Composition of the Oxide Film

The morphology of the formed oxide film (Figure 4) was further studied to elucidate the oxidation mechanisms of the alloys. For AlCoCrNiNb_0.2_ HEA, the surface of the formed oxide layer is quite dense, without cracks, pores, and chips at both 1000 °C and 1100 °C. In contrast, the oxide layer on AlCoCr0.5NiNb_0.2_ displayed a heterogeneous structure, comprising both dense and porous regions. The proportion of porous areas increased markedly with rising temperature.

Energy-dispersive X-ray spectroscopy (EDS) analysis (Table 3) revealed that the oxide films on both alloys were predominantly composed of aluminum, chromium, and oxygen. However, in the alloy with lower chromium content (AlCoCr_0.5_NiNb_0.2_), the chromium concentration within the oxide was correspondingly lower, while niobium content was significantly higher. Notably, the average niobium concentration in the oxide layer more than doubled with increasing temperature. This trend is further supported by a compositional analysis of a typical porous region in the surface oxide layer of AlCoCr_0.5_NiNb_0.2_ after 100 h of isothermal exposure at 1100 °C (Figure 5). These regions consist of crystal clusters enriched in oxygen and niobium, which fail to coalesce into a continuous, dense compact layer, instead forming a porous microstructure characteristic of the “loose areas” observed. The diminished high-temperature oxidation resistance of AlCoCr_0.5_NiNb_0.2_ is likely attributed to the formation of such porous regions. The reduction in chromium content appears to enhance the involvement of niobium in the oxidation process, thereby promoting the development of niobium-rich oxides’ “loose areas”.

Furthermore, the formation of porous zones may be associated with phase transitions of oxides when the temperature changes, for example, upon cooling. So, Nb_2_O_5_ undergoes a number of phase transformations [38]. Such processes are associated with the restructuring of the crystal lattice, which is accompanied by changes in the size of the crystallites and lattice volume, and can cause crack formation.

The cross-sectional analysis of the oxidized samples (Figure 6 and Figure 7) is consistent with the trends observed in the oxidation kinetics (Figure 3). For the AlCoCrNiNb_0.2_ HEA, the oxide film’s thickness (according to at least 10 measurements on different areas of the transverse section) was estimated to be approximately 3 ± 2 μm after oxidation at 1000 °C and 6 ± 2 μm at 1100 °C. In contrast, the AlCoCr_0.5_NiNb_0.2_ alloy exhibited lower oxidation resistance, as evidenced by a greater specific weight gain at both temperatures (Figure 3b), which corresponded to thicker oxide layers of approximately 7 ± 2 μm at 1000 °C and 13 ± 3 μm at 1100 °C. Furthermore, the AlCoCr_0.5_NiNb_0.2_ alloy demonstrated poorer adhesion between the oxide layer and the underlying metallic matrix. This is evident from microstructural observations of the cross-sectional samples, where cracks and pores were frequently observed at the oxide–metal interface (Figure 7), indicating a less stable and less protective oxide layer compared to that formed on AlCoCrNiNb_0.2_. Cross-sectional analysis also revealed the multilayered nature of the oxide films. Distinct regions enriched in aluminum and chromium were observed, while areas enriched in niobium became increasingly prominent with increasing oxidation temperature.

The XRD patterns of the alloys’ surfaces after high-temperature oxidation are presented in Figure 8. For both HEA compositions, aluminum oxide (Al_2_O_3_) was identified as the primary phase in the oxide film. Additionally, the formation of chromium oxide (Cr_2_O_3_) and niobium oxide (Nb_2_O_5_) was observed on the surfaces of both alloys. However, it was observed that chromium content significantly influences the phase composition of the oxide layers. In the AlCoCrNiNb_0.2_ alloy, spinel phases such as CoCr_2_O_4_ and NiCr_2_O_4_ were detected at both 1000 °C and 1100 °C, whereas these phases were absent in the AlCoCr_0.5_NiNb_0.2_ alloy. Furthermore, the formation of aluminum niobate (AlNbO_4_) was only observed at 1100 °C in the AlCoCrNiNb_0.2_ sample, but it was already present at 1000 °C in the AlCoCr_0.5_NiNb_0.2_ alloy. These findings are consistent with the previously discussed surface morphology and compositional analyses, reinforcing the conclusion that a reduction in chromium content leads to increased participation of niobium in the high-temperature oxidation process.

### 3.4. Oxidation Mechanism

At the initial stage of oxidation, which occurs in a kinetic mode, the composition of the alloys plays a fundamental role. In this case, the rate of diffusion of metal atoms to the surface (for interaction with oxygen atoms adsorbed on the surface) is the decisive factor. Aluminum has a small atomic radius and a high diffusion rate at elevated temperatures (including in high-entropy alloys) [39,40,41]. Chromium is also characterized by a high diffusion rate that ensures the formation of a protective film on the surface of the alloys [41,42]. The diffusion rate of niobium cations is lower than that of other metals in the composition of the investigated HEAs [21]. However, it is necessary to take into account the thermodynamics of the oxidation process. According to the Ellingham diagram [43], the most preferable and first oxide formation is the formation of aluminum oxide Al_2_O_3_, then the formation of Nb_2_O_5_ and Cr_2_O_3_, and then cobalt and nickel oxides.

Thus, it can be noted that at sufficiently high concentrations of Al and Cr (Al/Cr = 1:1) in the composition of the HEAs, oxidation products of niobium are found in a lesser extent than aluminum and chromium oxides. A decrease in the concentration of chromium in the alloy composition leads to a greater contribution of niobium to high-temperature oxidation and the formation of a greater amount of niobium oxidation products.

Further oxidation of the studied samples is subjected to the diffusion oxidation mode (it is the dominant one for these HEAs). The limiting stage in the diffusion mode is the diffusion rate of metal/oxygen atoms through the formed oxide film. Here, the formation of a continuous, defect-free film on the surface that prevents diffusion plays a decisive role [44]. The presence of “loose zones” can facilitate the penetration of oxygen ions through the surface film to the metal matrix, thereby facilitating and accelerating the oxidation process. Therefore, the AlCoCr_0.5_NiNb_0.2_ alloy showed worse characteristics in terms of resistance to high-temperature oxidation.

The microstructure of the samples also contributes to their behavior during high-temperature oxidation. The formation of Laves phases can increase the probability of the formation of niobium-containing oxide compounds. The alternation of solid solution and Laves phases at the alloy’s surface can compensate for the low diffusion rate of niobium cations at elevated temperatures.

The diagrams of oxidation of the studied alloys are shown in Figure 9.

The results obtained in this study are confirmed by earlier works [45,46], which indicate that the role of the alloying elements introduced into the composition of the HEAs cannot be underestimated; sometimes the alloying elements play a decisive role in the formation of the oxide film and its protective characteristics.

## 4. Conclusions

In summary, the following conclusions can be drawn from the study:

(1) For both studied alloys, the precipitation of niobium in the form of Laves phases along the grain boundaries of the BCC solid solution is characteristic.

(2) The AlCoCrNiNb_0.2_ HEA demonstrates greater resistance to high-temperature oxidation compared to the AlCoCr_0.5_NiNb_0.2_ alloy. A decrease in chromium concentration in HEAs leads to a decrease in the adhesion of the formed surface oxide layer to the metal matrix. Moreover, the morphology of the oxide film in AlCoCr_0.5_NiNb_0.2_ HEA is looser and more porous compared to the dense oxide film formed on the surface of the AlCoCrNiNb_0.2_ alloy.

(3) Niobium takes part in the high-temperature oxidation process with the formation of niobium oxide Nb_2_O_5_ and AlNbO_4_ and CrNbO4 compounds, which are inferior in protective qualities to aluminum oxide Al_2_O_3_. With a decrease in the Cr content in HEAs, niobium begins to play a more leading role in the oxidation of the alloy and the formation of the surface oxide layer.

(4) The obtained oxidation kinetic data for AlCoCrNiNb_0.2_ HEA indicated that the oxidation rate of this alloy is lower than that of stainless steel and a number of Fe-containing HEAs.

(5) Finally, it should be stated that the AlCoCrNiNb_0.2_ HEA is potentially suitable for metallic bond coat in thermal barrier coatings. However, further studies should be directed toward cyclic oxidation at higher temperatures up to 1200 °C. Additive manufacturing of AlCoCrNiNb_0.2_ coatings deposited by laser cladding and the study of their oxidation behaviors can be another direction for future works.

## Figures and Tables

**Figure 1 materials-18-03701-f001:**
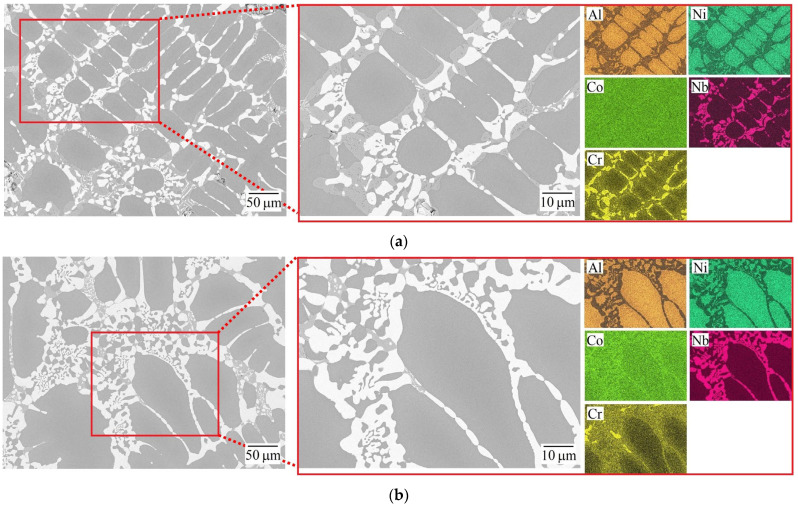
SEM (back-scattered electrons mode) micrographs and the corresponding EDS maps of the as-cast HEAs. (**a**) AlCoCrNiNb_0.2_ and (**b**) AlCoCr_0.5_NiNb_0.2_.

**Figure 2 materials-18-03701-f002:**
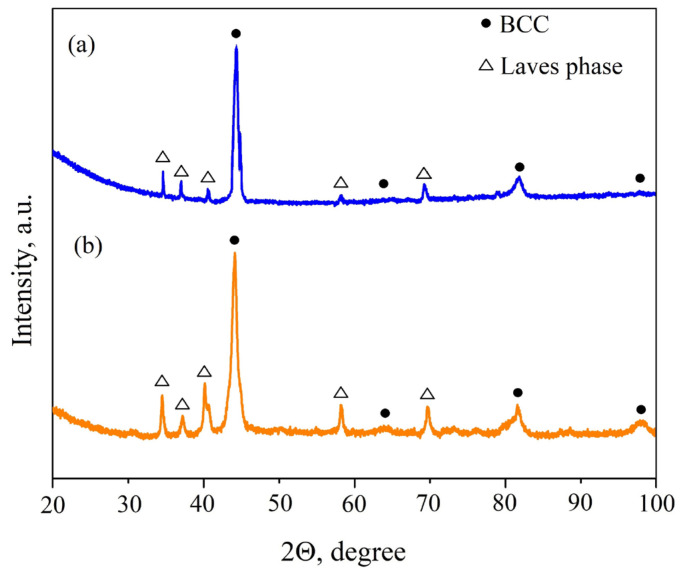
XRD patterns of the as-cast (**a**) AlCoCrNiNb_0.2_ and (**b**) AlCoCr_0.5_NiNb_0.2_ HEAs.

**Figure 3 materials-18-03701-f003:**
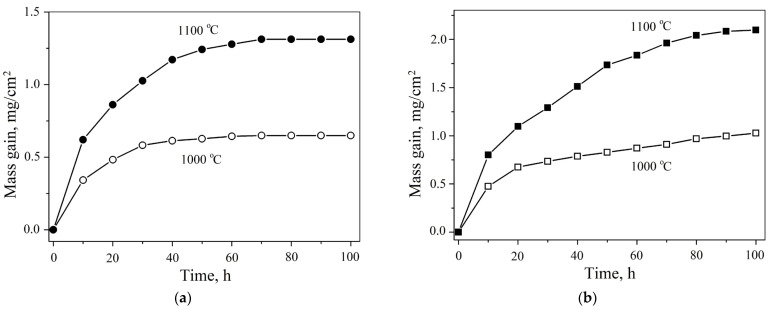
Kinetic curves of oxidation for (**a**) AlCoCrNiNb_0.2_ and (**b**) AlCoCr_0.5_NiNb_0.2_ HEAs.

**Figure 4 materials-18-03701-f004:**
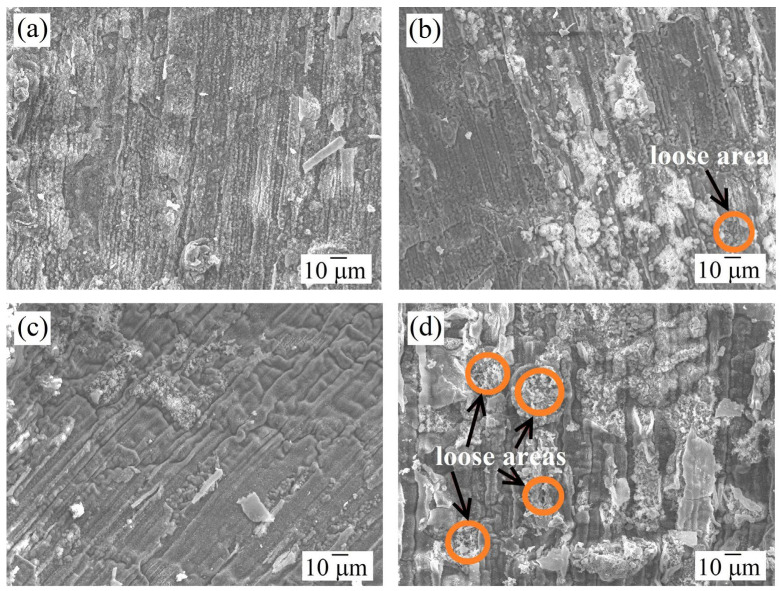
Surface morphology of the formed film at (**a**,**b**) 1000 °C and (**c**,**d**) 1100 °C for (**a**,**c**) AlCoCrNiNb_0.2_ and (**b**,**d**) AlCoCr_0.5_NiNb_0.2_ samples.

**Figure 5 materials-18-03701-f005:**
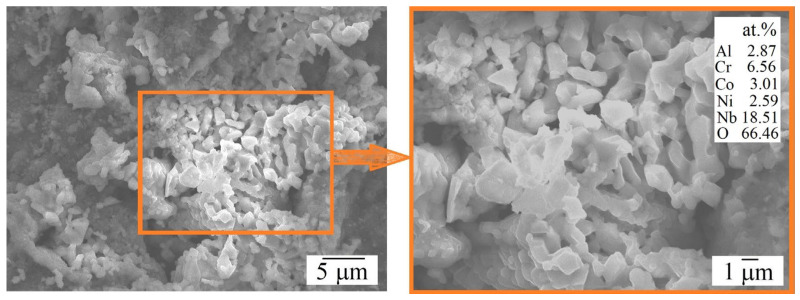
Morphology and composition of a typical “loose area” formed on the surface of AlCoCr_0.5_NiNb_0.2_ HEA after 100 h of exposure at 1100 °C.

**Figure 6 materials-18-03701-f006:**
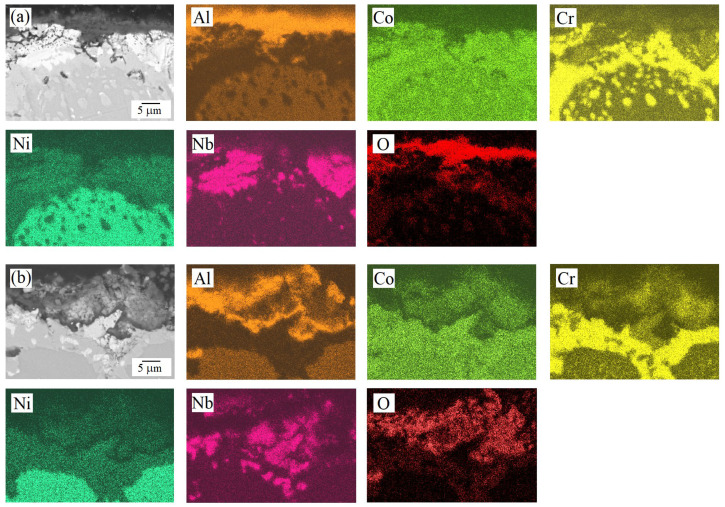
SEM (back-scattered electrons mode) micrographs and the corresponding EDS maps of the transverse section of AlCoCrNiNb_0.2_ HEA samples after 100 h of exposure at (**a**) 1000 °C and (**b**) 1100 °C.

**Figure 7 materials-18-03701-f007:**
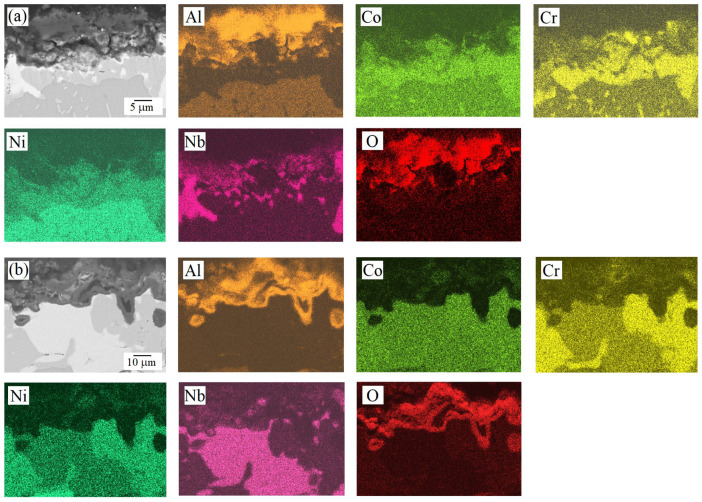
SEM (back-scattered electrons mode) micrographs and the corresponding EDS maps of the transverse section of AlCoCr_0.5_NiNb_0.2_ HEA samples after 100 h of exposure at (**a**) 1000 °C and (**b**) 1100 °C.

**Figure 8 materials-18-03701-f008:**
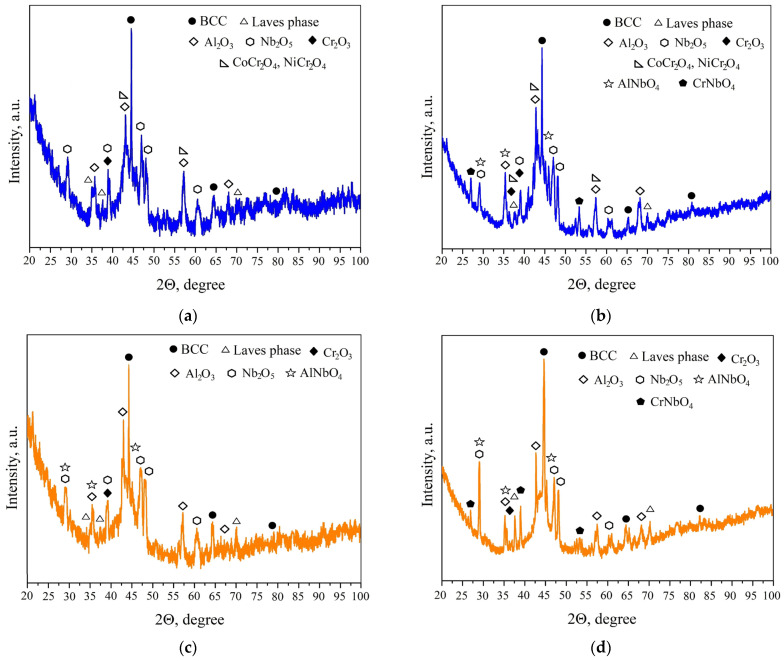
XRD patterns of the HEA samples’ surface. (**a**,**b**) AlCoCrNiNb_0.2_ and (**c**,**d**) AlCoCr_0.5_NiNb_0.2_ after 100 h of isothermal holding at (**a**,**c**) 1000 °C and (**b**,**d**) 1100 °C.

**Figure 9 materials-18-03701-f009:**
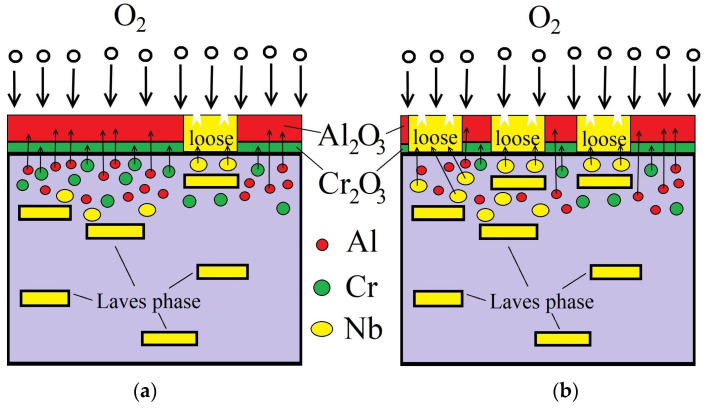
Schematic diagram of sample oxidation: (**a**) AlCoCrNiNb_0.2_ and (**b**) AlCoCr_0.5_NiNb_0.2_.

**Table 1 materials-18-03701-t001:** Chemical composition (EDS data, at.%) of the as-cast HEAs. Av—average composition; D—dendrite; ID—interdendritic region; MC—microstructural components.

HEA	MC	Al	Co	Cr	Ni	Nb
AlCoCrNiNb_0.2_	Av	23.76	23.87	23.66	23.85	4.86
	D	33.78	20.02	12.62	31.65	1.93
	ID1	8.41	25.26	49.78	15.26	1.29
	ID2	4.26	30.53	26.60	14.14	24.47
AlCoCr_0.5_NiNb_0.2_	Av	27.15	26.91	13.05	26.88	6.01
	D	38.52	20.38	6.21	32.85	2.04
	ID1	9.82	33.78	30.06	23.79	2.55
	ID2	4.42	35.96	17.65	15.47	26.50

**Table 2 materials-18-03701-t002:** Oxidation rate parabolic constant (*k*_p_, g^2^/cm^4^s) and activation energy (*E*_a_, kJ/mole) compared with the literature data.

Alloy	*t*, °C	*k* _p_	*E* _a_	Alloy	*t*, °C	*k* _p_	*E* _a_
AlCoCrNiNb_0.2_	1000	19.33 × 10^−13^	192.8	Al_0.5_CoCrFeNiCuPt_0.3_ [26]	900	1.09 × 10^−13^	170
	1100	72.87 × 10^−13^			1000	4.29 × 10^−13^	
AlCoCr_0.5_NiNb_0.2_	1000	41.79 × 10^−13^	184.6	Al_0.5_CoCrFeNi [31]	900	8.91 × 10^−13^	125.7
	1100	148.83 × 10^−13^			1000	39.3 × 10^−13^	
Al_30_(CoCrFeNi)_70_ [11]	1050	19 × 10^−13^	–	Al_0.6_CrFeCoNi [34]	900	11.8 × 10^−13^	196
Al_8_(CoCrFeNi)_92_ [11]	1050	25 × 10^−13^	–		1000	57.2 × 10^−13^	
CoCrFeNi [21]	1000	231 × 10^−13^	–	AISI 304L [36]	900	23 × 10^−13^	174
CoCrFeNiMo_0.2_ [21]	1000	161 × 10^−13^	–		1050	93 × 10^−13^	
CoCrFeNiMo_0.2_Nb_0.1_ [21]	1000	125 × 10^−13^	–	N5–3Ta nickel-based superalloy [37]	1050	0.61 × 10^−13^	180
CoCrFeNiMo_0.2_Nb_0.2_ [21]	1000	63.9 × 10^−13^	–		1150	1.93 × 10^−13^	

**Table 3 materials-18-03701-t003:** Average chemical composition of the oxide film determined by EDS analysis (at.%).

HEA	*t*, °C	Al	Co	Cr	Ni	Nb	O
AlCoCrNiNb_0.2_	1000	32.30	2.07	4.90	1.13	1.23	58.37
	1100	28.81	2.30	6.91	0.94	1.63	59.41
AlCoCr_0.5_NiNb_0.2_	1000	31.15	3.15	3.19	2.20	2.07	58.24
	1100	26.97	2.80	3.02	2.15	4.87	60.19

## Data Availability

The original contributions presented in this study are included in the article. Further inquiries can be directed to the corresponding author.

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
