# Peer review of "High-Temperature Oxidation Resistance of Fe-Free AlCoCrNiNb0.2 and AlCoCr0.5NiNb0.2 High-Entropy Alloys"

_materials, 2025, doi:10.3390/ma18153701_

Round 1

Reviewer 1 Report

Comments and Suggestions for Authors

This manuscript investigates the high-temperature oxidation behavior of Fe-free AlCoCrNiNb0.2 and AlCoCr0.5NiNb0.2 high-entropy alloys (HEAs). The experimental design is sound, and the analysis is thorough. It provides significant theoretical insights for the high-temperature applications of HEAs and demonstrates a certain level of innovation. Overall, manuscript is good, it can be accepted after doing this major revision carefully.

  1. The introduction provides an overview of the high-temperature oxidation behavior of high-entropy alloys (HEAs), but it does not discuss the impact of ionic diffusion mechanisms on oxidation kinetics. It is suggested to refer to the relevant studies to further explore the role of ionic diffusion in the high-temperature oxidation of HEAs.
  2. The article mentions that the phases of the metal and the oxide film were determined by XRD, but the specific parameters of the XRD analysis (such as scanning step size, scanning time, etc.) are not detailed. It is suggested to supplement these experimental parameters so that other researchers can reproduce the experiment.
  3. The selection of high-temperature oxidation temperatures is a crucial aspect of this study. However, the rationale behind choosing the specific temperatures of 1000°C and 1100°C for the oxidation experiments is not clearly explained. How do these temperatures relate to the practical application scenarios or the intrinsic material properties of the AlCoCrNiNb0.2 and AlCoCr0.5NiNb0.2 high-entropy alloys? A more detailed justification for the temperature selection, including any considerations of the alloy's phase transformation temperatures, service temperature ranges, or other relevant factors, would enhance the clarity and applicability of the research.
  4. The experimental section provides a detailed description of the alloy preparation and oxidation process. However, when discussing the mechanism of high-temperature oxidation, It is suggested to refer to the relevant studies to further explore the diffusion behavior of elements and the formation mechanism of oxide scales.
  5. When analyzing the morphology of the oxide film, the “loose areas” and other characteristic regions on the surface of the oxide film are only analyzed for their composition through EDS, but the physical processes during their formation, such as stress changes and phase transformations, are not discussed. It is suggested to supplement the relevant analysis.
  6. When discussing oxidation kinetics, although the oxidation rate constants of different alloys are compared, the underlying microscopic mechanisms behind these data are not elaborated in depth. For example, how the differences in alloy composition specifically affect the ion diffusion and other microscopic processes can be further supplemented.
  7. The manuscript clearly summarizes the high-temperature oxidation behavior of the two alloys. When discussing the differences in oxidation resistance between the alloys, you could reference the research to further emphasize the impact of alloy composition on oxidation behavior.

Author Response

Dear Reviewer, first of all we would like to thank you for your constructive comments, which helped us to improve the manuscript and will be useful for our future work.

All corrections in the text of the manuscript are highlighted in yellow.

Reviewer 1:

This manuscript investigates the high-temperature oxidation behavior of Fe-free AlCoCrNiNb0.2 and AlCoCr0.5NiNb0.2 high-entropy alloys (HEAs). The experimental design is sound, and the analysis is thorough. It provides significant theoretical insights for the high-temperature applications of HEAs and demonstrates a certain level of innovation. Overall, manuscript is good, it can be accepted after doing this major revision carefully.

  1. The introduction provides an overview of the high-temperature oxidation behavior of high-entropy alloys (HEAs), but it does not discuss the impact of ionic diffusion mechanisms on oxidation kinetics. It is suggested to refer to the relevant studies to further explore the role of ionic diffusion in the high-temperature oxidation of HEAs.

Answer: The corresponding changes have been made to the text of the manuscript.

“The positive effect of yttrium and platinum is primarily due to their influence on the diffusion processes occurring during oxidation. Introducing these elements provokes the formation of an aluminum oxide protective film, while it slows down the oxidation processes of other components in the alloy.”

2. The article mentions that the phases of the metal and the oxide film were determined by XRD, but the specific parameters of the XRD analysis (such as scanning step size, scanning time, etc.) are not detailed. It is suggested to supplement these experimental parameters so that other researchers can reproduce the experiment.

Answer: The corresponding changes have been made to the text of the manuscript.

“X-ray diffraction (XRD) was carried out on a Rigaku Ultima IV X-ray diffractometer (Rigaku, Tokyo, Japan) using Cu-Kα radiation (λ = 0.15406 Å) with the following shooting parameters: scanning range varied from 5° to 100°, step size was equal to 0.02°, scanning rate was 5 degrees per minute.”

3. The selection of high-temperature oxidation temperatures is a crucial aspect of this study. However, the rationale behind choosing the specific temperatures of 1000°C and 1100°C for the oxidation experiments is not clearly explained. How do these temperatures relate to the practical application scenarios or the intrinsic material properties of the AlCoCrNiNb0.2 and AlCoCr0.5NiNb0.2 high-entropy alloys? A more detailed justification for the temperature selection, including any considerations of the alloy's phase transformation temperatures, service temperature ranges, or other relevant factors, would enhance the clarity and applicability of the research.

Answer: The corresponding changes have been made to the text of the manuscript.

“The test temperatures were selected based on the presumption that these Nb-containing HEAs could be good candidates for metallic bond coat in thermal barrier coatings as a replacement for nickel superalloys. The use of HEAs as a bond coat implies working temperatures in the range of 1000 °C and above, reaching 1100 °C and even 1150-1200 °C.”

4. The experimental section provides a detailed description of the alloy preparation and oxidation process. However, when discussing the mechanism of high-temperature oxidation, It is suggested to refer to the relevant studies to further explore the diffusion behavior of elements and the formation mechanism of oxide scales.

Answer: The corresponding changes have been made to the text of the manuscript.

“Considering the observed differences in the oxidation behavior of the samples, while the manufacturing process and experimental conditions were the same, it can be inferred that high-temperature oxidation resistance of the alloys is mainly dominated by their chemical composition. The difference in chromium content can affect the kinetic processes associated with the diffusion of ions during oxidation. Sabioni et al. [38] showed that chromium has a significant effect on the high temperature phenomena taking place, since it has a sufficiently high diffusion rate to form a protective film. Therefore, a decrease in its concentration will contribute to a decrease in resistance against high-temperature oxidation.”

[38]. Sabioni, A.C.S.; Souza, J.N.V.; Ji, V.; Jomard, F.; Trindade, V.B.; Carneiro, J.F. Study of ion diffusion in oxidation films grown on a model Fe–15%Cr alloy. Solid State Ionics 2015, 276, 1–8. Doi: 10.1016/j.ssi.2015.03.027.

“The formation of loose zones can facilitate the penetration of oxygen ions through the surface film to the metal matrix, thereby facilitating and accelerating the oxidation process. To ensure high-temperature resistance of alloys, both the diffusion rate of ions from the metal matrix to the interface and the diffusion rate of oxygen ions through the surface film are important [39]. Therefore, the morphological features of the surface layer are very important when discussing high-temperature oxidation behavior of HEAs.”

[39]. Sabioni, A.C.S.; Malheiros, E.A.; Ji, V.; Jomard, F.; de Almeida Macedo, W.A.; Gastelois, P.L. Ion diffusion study in the oxide layers due to oxidation of AISI 439 ferritic stainless steel. Oxid. Met. 2014, 81, 407–419. Doi: 10.1007/s11085-013-9451-6.

5. When analyzing the morphology of the oxide film, the “loose areas” and other characteristic regions on the surface of the oxide film are only analyzed for their composition through EDS, but the physical processes during their formation, such as stress changes and phase transformations, are not discussed. It is suggested to supplement the relevant analysis.

Answer: The corresponding changes have been made to the text of the manuscript.

“Furthermore, the formation of porous zones may be associated with phase transitions of oxides when the temperature changes, for example, upon cooling. So, Nb2O5 undergoes a number of phase transformations [40]. Such processes are associated with the restructuring of the crystal lattice, which accompanied by changes in the size of the crystallites and lattice volume, and can cause crack formation.”

[40]. de M. Gomes, G.H.; de Andrade, R.R.; Mohallem, N.D.S. Investigation of phase transition employing strain mapping in TT- and T-Nb2O5 obtained by HRTEM micrographs. Micron 2021, 148, 103112. Doi: 10.1016/j.micron.2021.103112.

6. When discussing oxidation kinetics, although the oxidation rate constants of different alloys are compared, the underlying microscopic mechanisms behind these data are not elaborated in depth. For example, how the differences in alloy composition specifically affect the ion diffusion and other microscopic processes can be further supplemented.

Answer: The corresponding changes have been made to the text of the manuscript.

“Considering the observed differences in the oxidation behavior of the samples, while the manufacturing process and experimental conditions were the same, it can be inferred that high-temperature oxidation resistance of the alloys is mainly dominated by their chemical composition. The difference in chromium content can affect the kinetic processes associated with the diffusion of ions during oxidation. Sabioni et al. [38] showed that chromium has a significant effect on the high temperature phenomena taking place, since it has a sufficiently high diffusion rate to form a protective film. Therefore, a decrease in its concentration will contribute to a decrease in resistance against high-temperature oxidation.”

[38]. Sabioni, A.C.S.; Souza, J.N.V.; Ji, V.; Jomard, F.; Trindade, V.B.; Carneiro, J.F. Study of ion diffusion in oxidation films grown on a model Fe–15%Cr alloy. Solid State Ionics 2015, 276, 1–8. Doi: 10.1016/j.ssi.2015.03.027.

“The formation of loose zones can facilitate the penetration of oxygen ions through the surface film to the metal matrix, thereby facilitating and accelerating the oxidation process. To ensure high-temperature resistance of alloys, both the diffusion rate of ions from the metal matrix to the interface and the diffusion rate of oxygen ions through the surface film are important [39]. Therefore, the morphological features of the surface layer are very important when discussing high-temperature oxidation behavior of HEAs.”

[39]. Sabioni, A.C.S.; Malheiros, E.A.; Ji, V.; Jomard, F.; de Almeida Macedo, W.A.; Gastelois, P.L. Ion diffusion study in the oxide layers due to oxidation of AISI 439 ferritic stainless steel. Oxid. Met. 2014, 81, 407–419. Doi: 10.1007/s11085-013-9451-6.

7. The manuscript clearly summarizes the high-temperature oxidation behavior of the two alloys. When discussing the differences in oxidation resistance between the alloys, you could reference the research to further emphasize the impact of alloy composition on oxidation behavior.

Answer: The corresponding changes have been made to the text of the manuscript.

“The results obtained in this study are confirmed by earlier work [41], which indicate that the role of the alloying elements introduced into the composition of the HEAs cannot be underestimated; sometimes it plays a decisive role in the formation of the oxide film and its protective characteristics. In addition, the Al/Cr ratio can make a contribution to high-temperature oxidation resistance of HEAs [42].”

[41]. Shaburova, N.A.; Ostovari Moghaddam, A.; Veselkov, S.N.; Sudarikov, M.V.; Samoilova, O.V.; Trofimov, E.A. High-temperature oxidation behavior of AlxCoCrFeNiM (M = Cu, Ti, V) high-entropy alloys. Phys. Mesomech. 2021, 24(6), 653–662. Doi: 10.1134/S1029959921060035.

[42]. Sun, X.; Li, X.; Guo, S.; Zhu, L.; Teng, J.; Jiang, L.; Moverare, J.; Li, X.-H.; Peng, R.L. The impact of Al/Cr ratio on the oxidation kinetics of Y-doped AlCoCrFeNi high-entropy alloys at 1100 °C. Intermetallics 2025, 176, 108582. Doi: 10.1016/j.intermet.2024.108582.

Reviewer 2 Report

Comments and Suggestions for Authors

This study investigates the oxidation resistance of Fe-free AlCoCrNiNb high-entropy alloys at elevated temperatures. The work is interesting and relevant; however, in my opinion, several major comments should be carefully addressed to strengthen the manuscript and ensure its reproducibility and clarity.

1) In my opinion, the term “interdendritic space” is not very precise scientifically, and I would suggest using the term “interdendritic region” instead.

2) Please specify the type and manufacturer of the vacuum furnace.

3) Were the oxidation tests conducted according to any recognized standard (e.g., ASTM, ISO)? If not, I recommend specifying how the testing parameters were established.

4) It would also be helpful if the authors explained the rationale for selecting 1000 °C and 1100 °C as oxidation test temperatures. Were these chosen about a specific application or for comparability with previous studies?

5) The manuscript does not specify the number of samples tested for each alloy in the oxidation experiments. Was each composition evaluated on a single sample, or are the reported values based on averages from multiple measurements? Please clarify this point and indicate whether Figure 3 represents data from a single sample or averaged results, and if applicable, include standard deviations or error bars to reflect data variability

6) The Materials and Methods section lacks information on the sample preparation for SEM/EDS analysis. Please describe the metallographic procedure.

7) The findings in lines 150–158 would be clearer if presented in the form of a table for better readability.

8) In Figure 4, please indicate the porous and dense areas (e.g., with arrows) and clarify on what basis you claim that the proportion of porous regions increased significantly with temperature. Without quantification or marking, this statement appears subjective.

9) Could the authors provide quantitative measurements of the oxide layer thickness (e.g., average values from multiple areas)?

10) For better readability, the authors could consider separating the Results and Discussion sections into two distinct chapters.

11) The authors refer to the investigated compositions as “newly developed”; however, similar Fe-free AlCoCrNiNb HEA systems have already been studied (e.g., Pan et al. [24], Wu et al. [21], Jiang et al. [27]). I recommend that the authors provide a clearer explanation of how their specific alloys and study differ from previous works, whether this difference lies only in the Cr/Nb concentration or also in processing, experimental design, or potential applications.

12) The conclusion is brief and primarily reiterates the results. I recommend adding a short discussion on the practical significance of these findings (e.g., potential applications of the studied alloys) and any limitations of the study, along with suggestions for future research (e.g., cyclic oxidation, longer exposure times).

Author Response

Dear Reviewer, first of all we would like to thank you for your constructive comments, which helped us to improve the manuscript and will be useful for our future work.

All corrections in the text of the manuscript are highlighted in green.

Reviewer 2:

This study investigates the oxidation resistance of Fe-free AlCoCrNiNb high-entropy alloys at elevated temperatures. The work is interesting and relevant; however, in my opinion, several major comments should be carefully addressed to strengthen the manuscript and ensure its reproducibility and clarity.

1) In my opinion, the term “interdendritic space” is not very precise scientifically, and I would suggest using the term “interdendritic region” instead.

Answer: The corresponding changes have been made to the text of the manuscript.

2) Please specify the type and manufacturer of the vacuum furnace.

Answer: The corresponding changes have been made to the text of the manuscript.

“The HEAs ingots were produced by melting granules and powders of high-purity metals (> 99.9 wt. %) using a vacuum furnace NABERTHERM VHT 8/22-GR (Nabertherm GmbH, Lilienthal, Germany).”

3) Were the oxidation tests conducted according to any recognized standard (e.g., ASTM, ISO)? If not, I recommend specifying how the testing parameters were established.

Answer: The testing procedure was described in detail in our previous work [Samoilova, O.; Suleymanova, I.; Shaburova, N.; Ostovari Moghaddam, A.; Trofimov, E. The behavior of Al0.5CoCrFeNiCuPt0.3 high-entropy alloy during high-temperature oxidation. High Temp. Corr. Mater. 2024, 101, 811–825. Doi: 10.1007/s11085-024-10248-9].

Therefore, we cannot repeat ourselves in the description of the experiment; the editors will determine a large percentage of similarity index.

The editors have already asked us to cut the “Materials and Methods” section because they are not satisfied with the similarity index.

4) It would also be helpful if the authors explained the rationale for selecting 1000 °C and 1100 °C as oxidation test temperatures. Were these chosen about a specific application or for comparability with previous studies?

Answer: The corresponding changes have been made to the text of the manuscript.

“The test temperatures were selected based on the presumption that these Nb-containing HEAs could be good candidates for metallic bond coat in thermal barrier coatings as a replacement for nickel superalloys. The use of HEAs as a bond coat implies working temperatures in the range of 1000 °C and above, reaching 1100 °C and even 1150-1200 °C.”

5) The manuscript does not specify the number of samples tested for each alloy in the oxidation experiments. Was each composition evaluated on a single sample, or are the reported values based on averages from multiple measurements? Please clarify this point and indicate whether Figure 3 represents data from a single sample or averaged results, and if applicable, include standard deviations or error bars to reflect data variability.

Answer: The corresponding changes have been made to the text of the manuscript.

“The changes in specific weight gain as a function of isothermal holding time are shown in Figure 3, each point on the graphs represents the result of averaging three parallel experiments; the calculated errors did not exceed 5%.”

6) The Materials and Methods section lacks information on the sample preparation for SEM/EDS analysis. Please describe the metallographic procedure.

Answer: The corresponding changes have been made to the text of the manuscript.

“To study the microstructure before and after oxidation, as well as to determine the thickness of the formed oxide film, polished sections were made from the experimental samples. For this, a Delta AbrasiMet (Buehler, Leinfelden-Echterdingen, Germany) cutting machine, a SimpliMet 1000 (Buehler, Leinfelden-Echterdingen, Germany) pressing machine, and an EcoMet 250/AutoMet 250 (Buehler, Leinfelden-Echterdingen, Germany) grinding machine were used. Grinding was carried out sequentially on 250 μm, 75 μm, 25 μm, 9 μm sandpaper, and a 3 μm diamond suspension was used for final polishing.”

7) The findings in lines 150–158 would be clearer if presented in the form of a table for better readability.

Answer: The corresponding changes have been made to the text of the manuscript.

“The results of the calculations of the oxidation parameters are summarized in Table 2.”

Table 2. Oxidation rate parabolic constant (kp, g2/cm4s) and activation energy (Ea, kJ/mole) compared with literature data.

Alloy

t, °C

kp

Ea

Alloy

t, °C

kp

Ea

AlCoCrNiNb0.2

1000

19.33 × 10–13

192.8

Al0.5CoCrFeNi [31]

900

8.91 × 10–13

125.7

1100

72.87 × 10–13

1000

39.3 × 10–13

AlCoCr0.5NiNb0.2

1000

41.79 × 10–13

184.6

Al0.6CrFeCoNi [34]

900

11.8 × 10–13

196

1100

148.83 × 10–13

1000

57.2 × 10–13

Al30(CoCrFeNi)70 [11]

1050

19 × 10–13

AISI 304L [36]

900

23 × 10–13

174

Al8(CoCrFeNi)92 [11]

1050

25 × 10–13

1050

93 × 10–13

Al0.5CoCrFeNiCuPt0.3 [26]

900

1.09 × 10–13

170

N5-3Ta nickel-based
superalloy [37]

1050

0.61 × 10–13

180

1000

4.29 × 10–13

1150

1.93 × 10–13

8) In Figure 4, please indicate the porous and dense areas (e.g., with arrows) and clarify on what basis you claim that the proportion of porous regions increased significantly with temperature. Without quantification or marking, this statement appears subjective.

Answer: The corresponding changes have been made to the text of the manuscript.

Figure 4. Surface morphology of the formed film at (a,b) 1000 °C and (c,d) 1100 °C
for (a,c) AlCoCrNiNb0.2, and (b,d) AlCoCr0.5NiNb0.2 samples.

9) Could the authors provide quantitative measurements of the oxide layer thickness (e.g., average values from multiple areas)?

Answer: The corresponding changes have been made to the text of the manuscript.

“For the AlCoCrNiNb0.2 HEA, the oxide film thickness (according to at least 10 measurements on different areas of the transverse section) was estimated to be approximately 3 ± 2 μm after oxidation at 1000 °C and 6 ± 2 μm at 1100 °C. In contrast, AlCoCr0.5NiNb0.2 alloy exhibited lower oxidation resistance, as evidenced by a greater specific weight gain at both temperatures (Figure 3b), which corresponded to thicker oxide layers of approximately 7 ± 2 μm at 1000 °C and 13 ± 3 μm at 1100 °C.”

10) For better readability, the authors could consider separating the Results and Discussion sections into two distinct chapters.

Answer: Thank you for comment. However, in the present manuscript, separating the Results and Discussion sections can break the logical chain of reasoning; it is more convenient for perception to provide a figures and tables and do the analysis than to provide experimental data and then analyze the results a few pages later. The reader will have to return to the data, which is extremely inconvenient.

11) The authors refer to the investigated compositions as “newly developed”; however, similar Fe-free AlCoCrNiNb HEA systems have already been studied (e.g., Pan et al. [24], Wu et al. [21], Jiang et al. [27]). I recommend that the authors provide a clearer explanation of how their specific alloys and study differ from previous works, whether this difference lies only in the Cr/Nb concentration or also in processing, experimental design, or potential applications.

Answer: Pan et al. [24] and Jiang et al. [27] studied mechanical properties but not high-temperature oxidation. Pan et al. [24] also examine corrosion resistance in sodium chloride solution. Wu et al. [21] studied the effects of niobium on the classic Cantor alloy (without aluminum in the composition). In addition, in studies [21, 27] iron is included in the composition of the HEAs. Indeed, in [24] there is a completely different Cr/Nb ratio.

The corresponding changes have been made to the text of the manuscript.

“In contrast to the HEAs compositions studied elsewhere [24], the present study investigates alloys with different Al/Nb, Cr/Nb, and Al/Cr ratios.”

12) The conclusion is brief and primarily reiterates the results. I recommend adding a short discussion on the practical significance of these findings (e.g., potential applications of the studied alloys) and any limitations of the study, along with suggestions for future research (e.g., cyclic oxidation, longer exposure times).

Answer: The corresponding changes have been made to the text of the manuscript.

“Finally, it should be stated that AlCoCrNiNb0.2 HEA is potentially suitable for metallic bond coat in thermal barrier coatings. However, further studies should be directed towards cyclic oxidation at higher temperatures up to 1200 °C. Additive manufacturing of AlCoCrNiNb0.2 coatings deposited by laser cladding and the study of their oxidation behaviors can be another direction for future works.”

Round 2

Reviewer 1 Report

Comments and Suggestions for Authors

accept it.

Author Response

Dear Reviewer, Thank you for recommending the manuscript for publication.

Reviewer 2 Report

Comments and Suggestions for Authors

All of my comments have been adequately addressed, and I recommend the manuscript for publication in its current form. I respect the authors’ decision to keep the Results and Discussion as a single combined section.

Author Response

(The authors gave the same response as above.)
